# Conjunctival Fluid Secretion Impairment via CaCC-CFTR Dysfunction Is the Key Mechanism in Environmental Dry Eye

**DOI:** 10.3390/ijms232214399

**Published:** 2022-11-19

**Authors:** Jinyu Zhang, Limian Lin, Xiaomin Chen, Shuyi Wang, Yuan Wei, Wenliang Zhou, Shuangjian Yang, Shiyou Zhou

**Affiliations:** 1State Key Laboratory of Ophthalmology, Zhongshan Ophthalmic Center, Sun Yat-sen University, Guangzhou 510060, China; 2School of Life Sciences, Sun Yat-sen University, Guangzhou 510275, China; 3Guangdong Institute for Vision and Eye Research, Guangzhou 510060, China

**Keywords:** environmental dry eye disease, conjunctival fluid secretion, CFTR, CaCC

## Abstract

Dry eye disease (DED) is a multifactorial disease with an incidence of approximately 50% worldwide. DED seriously affects quality of life and work. The prevalence of environmental DED (eDED) ranges from 35 to 48%. Conjunctival fluid secretion dysfunction may be one of the major causes of DED. Notably, the Cl^–^ flux corresponds to the conjunctival fluid secretion and could be affected by ATP. Both the cystic fibrosis transmembrane conductance regulator (CFTR) and the Ca^2+^-activated Cl^–^ channel (CaCC) are Cl^–^ channels involved in epithelial fluid secretion. Conjunctival fluid secretion could be increased by activating P2Y_2_R (an ATP receptor) in DED. However, the role of the CaCC and CFTR channels regulated by P2Y_2_R in eDED remains unclear. In this study, we established a rabbit eDED model using a controlled drying system. A Ussing chamber was used to perform a conjunctival short-circuit current induced by ATP to evaluate the reactivity of the ion channels to the ATP. Our results revealed that eDED accompanied by conjunctival fluid secretion impairment was caused by a P2Y_2_R dysfunction, which is related to CaCC-CFTR signaling in the conjunctiva epithelium. Notably, the coupling effect of the ATP-induced CaCC-CFTR activation and intracellular Ca^2+^ may represent a promising therapeutic target for treating eDED.

## 1. Introduction

More than one billion people suffer from dry eye disease (DED) worldwide. Severe DED can cause visual impairment, which severely affects the quality of life of patients [1,2]. DED is a multifactorial disease of the ocular surface, characterized by a loss of homeostasis of the tear film [3]. The predisposing factors for DED include environmental risk, androgen deficiency, contact lenses, and aging [4]. A mild DED, environmental dry eye disease (eDED), has become increasingly common, with an incidence of 35–48% [1]. Notably, tear production has been shown to be significantly decreased in eDED patients [5]. In tear production, the secretory rate of the conjunctival fluid exceeds the basal tear production [6]. Conjunctival secretion has been shown to contribute approximately one-fourth of the total tear production [7], indicating that conjunctival secretion could play an important role in tear production. Whether conjunctival secretion is impaired in eDED and the role it plays in the pathogenesis of eDED are still unknown.

The secretion of water from conjunctival epithelial cells was evaluated using chloride ion (Cl^–^) fluid. The outflow of the Cl^–^ flux corresponded to the conjunctival fluid secretion [8]. The cystic fibrosis transmembrane conductance regulator (CFTR) is a cAMP-dependent Cl^–^ channel that facilitates fluid secretion in the apical side of the conjunctiva [9]. The CFTR agonist could increase the tear volume in clinical DED patients [8]. However, no change in the Cl^−^ concentration was found in patient tears with cystic fibrosis due to the CFTR mutation [10].

ATP receptors exist in various types of epithelial cells, including the conjunctiva epithelium. P2Y_2_R is one type of ATP receptor. P2Y_2_R agonists, such as diquafosol, UTP, and ATP, have been used to increase the tear production in DED patients and rabbits [6,11,12]. Notably, the activation of P2Y_2_R increased the intracellular calcium ion (Ca^2+^) [13], which has an obvious effect on the regulation of the Cl^−^ flux in the airway epithelium [14]. Moreover, the application of A23187 (a Ca^2+^ ionophore) apically increased the Cl^−^ flux by 58% in the rabbit conjunctiva epithelium [15]. A protein corresponding to the Ca^2+^-activated Cl^−^ channel (CaCC) was detected on the apical side of the conjunctiva [16]. These findings indicate that the CaCC signaling may be involved in conjunctival secretion. However, there are no direct experimental data on the effects of the CaCC on conjunctival secretion. In the kidney inner medullary collecting duct cells and human airway epithelia, the CaCC signaling works with the CFTR [17,18]. Whether P2Y_2_R contributes to regulating the conjunctival fluid secretion via the combination of the CaCC and CFTR has not been elucidated, especially in eDED.

In this study, the impairment and pathogenesis of conjunctival epithelium secretion in eDED was investigated using an eDED rabbit model. Our study demonstrated that the P2Y_2_R signaling dysfunction is a key factor in the impairment of conjunctival secretion by the coactivation of the CaCC and CFTR. Our research revealed that a new molecular mechanism underlies the fluid secreted from the conjunctival epithelium, indicating a potential therapeutic target for treating eDED.

## 2. Results

### 2.1. eDED Has Ocular Surface Injury and Conjunctival Fluid Secretion Dysfunction

To investigate an ocular surface injury and the tear secretion in eDED, a controlled drying system (CDS) was established. In this study, the disruptions of the conjunctiva and cornea epithelium were stained with lissamine green and fluorescein, respectively. The staining score was positively correlated with the staining area. Compared with the control group, the lissamine green score increased from day 7 to day 14 in the dry eye group (day 7: 0.15 ± 0.08 vs. 1.30 ± 0.15; day 14: 0.15 ± 0.08 vs. 1.90 ± 0.23, all n = 10–11, *p* < 0.001) (Figure 1a). Similarly, the fluorescein staining score significantly increased from day 7 to day 14 in the dry eye group (day 7: 1.00 ± 0.26 vs. 5.00 ± 0.49; day 14: 1.10 ± 0.18 vs. 7.27 ± 0.65, all n =10–11, *p* < 0.001) (Figure 1b). The fluid secretion from the conjunctiva is one of the most vital components of tear production. As expected, the tear production also declined from day 7 to day 14 in the dry eye group (9.45 ± 0.47 vs. 5.80 ± 0.39 mm at day 7; 9.63 ± 0.51 vs. 3.20± 0.24 mm at day 14; all n = 10–11, *p* < 0.001) (Figure 1c). Consistent with the above results, the ocular epithelium combined with the fluid reduction in the conjunctiva was impaired in eDED.

### 2.2. P2Y_2_R Plays a Key Role in eDED

Activating ATP receptors increases the intracellular Ca^2+^ and cAMP [19,20]. To confirm the changes in the intracellular Ca^2+^ in eDED, we digested the eDED conjunctival epithelium and loaded it with an intracellular calcium fluorescence probe (Fluo-3/AM). The results showed that the intensity of the intracellular Ca^2+^ was decreased in the eDED conjunctival epithelium (10.80 ± 1.02 vs. 4.85 ± 0.9 AU, n = 6, *p* < 0.001) (Figure 2a). In addition, the concentration of the cAMP in the conjunctiva epithelium was reduced in the dry eye group (5.08 ± 0.26 vs. 4.30 ± 0.21 nmol/mL, n = 9, *p* < 0.05) (Figure 2b). Subsequently, the ATP did not evoke an intracellular Ca^2+^ release of the same amplitude in the conjunctival epithelium in the dry eye group as in the control group (Figure 2c). The ATP-dependent Isc reflects the reactivity of the ion channels to the ATP, which plays a role via P2Y_2_R [13]. The conjunctival fluid secretion is closely related to the function of the anion channel [8]. Notably, the initial peak (first peak) and sustained peak (second peak) of the conjunctival epithelium Isc were markedly decreased after the ATP stimulation in the dry eye group (first peak: 27.19 ± 3.06 vs. 7.08 ± 1.79, n = 6, *p* < 0.001; second peak: 28.84 ± 2.92 vs. 6.32 ± 1.28, n = 6, *p* < 0.05) (Figure 3a). These results demonstrated that the ATP receptor dysfunction in the conjunctiva epithelium plays a key role in eDED.

To elucidate the relationship between the P2Y_2_R and Isc variation, a P2Y_2_R inhibitor (suramin) was applied before the conjunctival ATP stimulation. The results showed that treatment with the P2Y_2_R inhibitor led to a decrease of 50% in the ATP-induced conjunctival Isc (first peak: 24.06 ± 3.24 vs. 11.98 ± 2.34; second peak: 27.27 ± 2.50 vs. 5.14 ± 2.10, all n = 6–8, *p* < 0.01) (Figure 3b). The ATP-induced conjunctival Isc decreased by more than half in the P2Y_2_R-siRNA treatment group compared with the control group (first peak: 16.67 ± 2.10 vs. 6.79 ± 3.17; second peak:15.84 ± 2.60 vs. 6.50 ± 2.84, all n = 4–5, *p* < 0.05) (Figure 3c). The rabbit ocular surfaces were also impaired using the P2Y_2_R-siRNA eyedrop treatment (Figure 3d). These findings showed that the ATP-induced conjunctival Isc appropriately showed conjunctival fluid secretion. The ATP-induced conjunctival Isc deficiency and subsequent decrease in the conjunctival fluid secretion in eDED contributed to the conjunctival P2Y_2_R dysfunction.

### 2.3. ATP-Induced Conjunctival Fluid Secretion Is Related to the Anion Channel

Both the Cl^–^ flux and HCO_3_^−^ flux were secreted from the Cl^–^ channel of the cell membrane surface [21]. A conjunctiva buffer solution without Cl^−^ or HCO_3_^−^ was used to determine whether the ATP-induced conjunctival Isc variation is related to the exchange of Cl^–^ or/and HCO_3_^–^. Compared with the normal buffer solution, the ATP-induced conjunctival Isc response decreased by approximately 50% in the Cl^–^-free buffer (first peak: 20.93 ± 2.54 vs. 8.10 ± 2.41; second peak: 23.91 ± 2.31 vs. 8.91 ± 1.85, all n = 5–12, *p* < 0.01) (Figure 4a,b,e) and in the HCO_3_^–^-free buffer (first peak: 20.93 ± 2.54 vs. 4.82 ± 0.89; second peak: 23.91 ± 2.32 vs. 6.13 ± 0.92, all n = 6–12, *p* < 0.001) (Figure 4b,c,e). Moreover, the ATP-induced conjunctival Isc sharply declined by over 90% in the Cl^–^- and HCO_3_^–^-free buffer (first peak: 20.93 ± 2.54 vs. 2.44 ± 0.72; second peak: 23.91 ± 2.31 vs. 1.97 ± 0.59, all n = 9–12, *p* < 0.001) (Figure 4a,d,e). These results suggest that the combination of the Cl^−^ and HCO_3_^−^ fluxes contributed to the ATP-induced conjunctival fluid secretion.

### 2.4. CaCC-CFTR Pathway Participates in the ATP-Induced Conjunctival Fluid Secretion

The CFTR is a cAMP-dependent Cl^−^ channel that is essential for conjunctival fluid secretion [8]. To determine whether the CFTR affects the ATP-induced conjunctival Isc variation, a CFTR inhibitor (CFTRinh-172) and a specific cAMP inhibitor (H_89_) were applied before the conjunctival ATP simulation. The CFTR inhibitor reduced the second peak of the conjunctival Isc without affecting the first peak (first peak: 20.59 ± 2.19 vs. 17.04 ± 3.15, n = 6–14, *p* < 0.05; second peak: 22.16 ± 2.40 vs. 8.96 ± 0.98 n = 6–14, *p* < 0.01) (Figure 5a,b,f). A similar trend was observed with a specific cAMP inhibitor (first peak: 20.59 ± 2.19 vs. 13.54 ± 1.98, n = 6–14, *p* < 0.05; second peak: 22.16 ± 2.40 vs. 5.18 ± 0.79, n= 6–14, *p* < 0.001) (Figure 5c,f). Furthermore, an adenylyl cyclase (AC)-specific inhibitor (MDL12330A), which is upstream of the cAMP, was used in the same way, resulting in a similar outcome (first peak: 20.59 ± 2.19 vs. 24.10 ± 0.44, n = 5–14, *p* < 0.05; second peak: 22.16 ± 2.40 vs. 2.69 ± 0.32, n = 5–14, *p* < 0.001) (Figure 5d,f). These results demonstrated that the CFTR participates in the conjunctival fluid secretion.

Subsequently, we focused on another Cl^–^ channel, CaCC, which is a Ca^2+^-activated Cl^–^ channel on the cell membrane of the conjunctiva epithelium [16]. To investigate the role of the CaCC in the conjunctival Isc, a specific CaCC blocker (T16Ainh-A01) was applied before the conjunctival ATP simulation. Compared with the vehicle group, the CaCC-specific blocker induced a decrease in the ATP-induced conjunctival Isc (first peak: 20.59 ± 2.19 vs. 9.25 ± 2.85, n = 5–14, *p* < 0.05; second peak: 22.16 ± 2.40 vs. 10.95 ± 4.11, n = 5–14, *p* < 0.05) (Figure 5e,f). These results indicate that the ATP-induced conjunctival fluid secretion is dependent on the activation of the CaCC-CFTR pathway.

### 2.5. Intracellular Ca^2+^ Regulates the ATP-Induced Conjunctival Fluid Secretion

To investigate the role of the intracellular Ca^2+^ in the ATP-evoked conjunctival Isc, a PLC inhibitor (U73122), an IP3R inhibitor (2-APB), and a Ca^2+^ chelator (BAPTA/AM) were applied before the ATP simulation. The ATP-induced conjunctival Isc in the PLC inhibitor group decreased to one-half of that in the vehicle group (first peak: 20.34 ± 2.20 vs. 10.90 ± 1.41, n = 6–14, *p* < 0.05; second peak: 22.30 ± 2.26 vs. 10.49 ± 1.36, n= 6–14, *p* < 0.01) (Figure 6a,b,e). A same result was found in the treatment with the IP3R inhibitor (first peak: 0.34 ± 2.20 vs. 2.01 ± 0.43; second peak: 22.30 ± 2.26 vs. 6.31 ± 2.68, all n = 6–14, *p* < 0.001) (Figure 6a,c,e). The results revealed that the intracellular Ca^2+^ declined in eDED (Figure 2d) as well as several sources of cytosolic Ca^2+^ after the P2Y_2_R activation [22]. The ATP-induced conjunctival Isc response was initiated by treatment of the Ca^2+^ chelator (first peak: 20.34 ± 2.20 vs. 5.00 ± 1.91, n = 6–14, *p* < 0.01; second peak: 22.30 ± 2.26 vs. 15.80 ± 2.73, n = 6–14, *p* < 0.05) (Figure 6a,d,e). These results demonstrate that the intracellular Ca^2+^ is necessary for the ATP-induced conjunctival Isc and subsequent conjunctival fluid secretion.

## 3. Discussion

In this study, P2Y_2_R dysfunction combined with the decreased intracellular cAMP and cytosolic Ca^2+^ in the conjunctival epithelium was demonstrated in an eDED rabbit model. The P2Y_2_R dysfunction caused a conjunctival fluid secretion impairment, which was derived from the ATP through the CaCC-CFTR signaling. The release of the intracellular Ca^2+^ plays a core cross-pivotal role in the CaCC-CFTR signaling and is indispensable for the ATP-induced conjunctival fluid secretion. Therefore, the impairment of the conjunctival fluid secretion via the CaCC-CFTR pathway dysfunction could be a key mechanism in eDED.

The previous studies have shown that fluid in the conjunctiva epithelium is transported from the stromal (basolateral) side to the mucosal (apical) side. Conjunctival fluid secretion is dependent on the outflow of the anion flux [8]. There are various ion channels on the conjunctival epithelium, among which aquaporins, CFTR, ENaC, and CaCC are closely related to tear secretion [8,23]. Isc was found to present cellular ion channel reactivity on different activators. The ATP-induced Isc reflects the reactivity of the ion channels to the ATP [24]. As shown in this study, more than 90% of the ATP-induced Isc amplitude responses from the apical side of the conjunctival epithelium were produced by the anion secretion, suggesting that the ATP-induced Isc response could well represent the conjunctival fluid. The secretion of Cl^−^ from the conjunctival basal side to the apical side via an exchange with the sodium ions was demonstrated to account for approximately 60% of the Isc response [25,26]. Therefore, the outfluxes of HCO_3_^−^ and other anions could occupy more than one-third of the ATP-induced Isc response. A similar fluid secretion was also detected in the cortical collecting duct cells [27] and the caudal epididymal epithelium [28].

To date, the activation of the P2Y receptors has been considered to stimulate the fluid secretion across the conjunctiva [24]. The levels of the intracellular cAMP and Ca^2+^ are elevated after the P2Y receptor activation [19,20]. The intrinsic process of activating P2Y_2_R is interrelated with the increase in the PLC and the transient increase in the intracellular Ca^2+^ concentration [11], suggesting that the cytosolic Ca^2+^ may regulate the conjunctival fluid secretion. Moreover, a release of the intracellular Ca^2+^ by another way, but not the PLC pathway, could have the same impact [29]. These results indicate that P2Y_2_R is a key factor in the conjunctival fluid secretion. However, the concentration of the ATP in tears increased in dry eye patients [30]. The concurrent cellular apoptosis of the conjunctiva epithelium, which resulted in P2Y_2_R dysfunction, was believed to cause increased tear ATP. The results of the present study showed that the P2Y_2_R dysfunction with decreased intracellular cAMP and Ca^2+^ existed in the conjunctiva of eDED, suggesting that P2Y_2_R plays a key role in eDED. In contrast, silencing P2Y_2_R led to a dry eye condition, and the inhibition of P2Y_2_R effectively suppressed the ATP-induced conjunctival Isc.

The CFTR is a cAMP-dependent Cl^−^ channel existing in both the corneal and conjunctival epithelium, which increased the basal tear volume after activation [8]. The present study also demonstrated that the ATP-induced conjunctival Isc was significantly suppressed by applying a CFTR inhibitor. Hence, the CFTR contributed to the ATP-induced conjunctival fluid secretion. However, the intracellular cAMP in the conjunctiva was decreased in the eDED model, indicating that the CFTR signaling was impaired in the dry eye condition.

In this study, the CaCC inhibitor decreased the ATP-induced conjunctival Isc, indicating that the CaCC signaling may be the main contributor to the ATP-induced conjunctival fluid secretion. The CaCC, another Cl^−^ channel, exists in the apical surface of the conjunctiva epithelium [31]. The CaCC is a Ca^2+^ dependent Cl^–^ channel that transports the fluid secretion from the airway epithelium [32]. Our results showed that increased intracellular Ca^2+^ activated a transient opening of the CaCC and then a persistent CFTR channel for the Cl^−^ secretion from the conjunctival epithelial cells. This study is the first in the literature to verify the combination of the CaCC signaling and CFTR in the Cl^–^ outflow across the conjunctival epithelium.

Our results showed that the content of the intracellular Ca^2+^ in the conjunctiva epithelium was decreased in eDED and did not respond to the ATP activation. The level of the intracellular Ca^2+^ was shown to be closely related to the function of the epithelial secretion, such as in the airway epithelium [13], caudal epididymal epithelium [28], and conjunctiva epithelium [29]. The intracellular Ca^2+^ in the conjunctiva epithelium can be regulated via different mechanisms. The activation of the transient receptor potential melastatin 8 channel increased the intracellular Ca^2+^ and then conducted sense information to the higher brain center [33]. In this study, P2Y_2_R dysfunction was found in the conjunctiva epithelium in eDED, which may be an intrinsic cause of the decreased intracellular Ca^2+^. Except for P2Y_2_R, the imbalance in the intracellular Ca^2+^ could be affected by endoplasmic reticulum stress and a mitochondrial matrix Ca^2+^ overload, which triggered cell apoptosis [34,35]. Endoplasmic reticulum stress-induced ocular surface epithelial cell apoptosis was shown in the pathogenesis of DED. Thus, further work is needed to explore the detailed mechanism of intracellular Ca^2+^ in eDED.

## 4. Materials and Methods

### 4.1. Animals

Male New Zealand white rabbits weighing 2.0–2.5 kg (purchased from Guangdong Provincial Center for Animal Research, Guangzhou, China) were chosen for the study. The rabbits were quarantined and acclimatized for one week before the experiments. All experimental procedures were approved by the Institutional Animal Care and Use Committee at Zhongshan Ophthalmic Center at Sun Yat-sen University and complied with all relevant ethical regulations for animal testing and research.

### 4.2. Environmental Dry Eye Disease Model

The rabbit eDED model was established using a controlled drying system (CDS). A detailed system for dry eye models was published in our previous report [36]. In brief, the rabbits in the dry eye group were reared in the CDS at a relative humidity of 22 ± 4%, an air flow of 3–4 m/s, and temperatures from 23 to 25 °C. The control group remained in a normal environment. Each rabbit was lissamine green/fluorescein stained as normal [9,37] and evaluated for tear production on days 0, 7, and 14.

### 4.3. Intervention of Rabbit Eyes with P2Y_2_R siRNA

Small interfering RNA (siRNA) sequences for silencing P2Y_2_R were applied topically four times per day to the ocular surfaces of the rabbits. The nucleotide sequence of the siRNA target site chosen to silence the expression of the rabbit P2Y_2_R-coding sequence (GenBank EU886321) was as follows: 5′-AAC CTG TAC TGC AGC ATC CTC-3′ [38,39]. The siRNA molecule was synthesized by RiboBio company (Guangzhou, China). The control group was given a normal saline as siRNA vehicle. The measurement of corneal fluorescein staining was in accordance with the above regulations.

### 4.4. Short-Circuit Current (Isc) Measurements

The Isc measurement was performed following the modified procedure described in our previous reports [40]. The superior conjunctival tissues were carefully isolated from the rabbits and clamped vertically between two halves of a Ussing chamber (EM-CSYS-2 Ussing Chamber Systems, Physiologic Instruments, San Diego, CA, USA) with an inner area of 0.45 cm^2^. The side of the epithelium was placed in a fixed direction facing the positive electrode. Then, the conjunctiva was bathed on both sides with bicarbonate Ringer’s solution composed of the following (in mM): 116.4 NaCl, 5.4 KCl, 5.4 NaHCO3, 0.78 NaH2PO4, 1.8 CaCl2, 0.81 MgCl2, 5.55 D-glucose, and 10 HEPES. It was then gassed with 95% O_2_/5% CO_2_ at 37 °C to maintain a pH of 7.4. The epithelium exhibited a basal transepithelial potential difference, which was measured by Ag/AgCl electrodes with KCl/Agar bridges connected to a voltage-clamp amplifier (VCC MC6, Physiologic Instruments, San Diego, CA, USA). The transepithelial potential difference in the conjunctiva epithelial layer was clamped at 0 mV for the Isc measurement.

The change in Isc was defined as the difference between the value at baseline and that at a peak following compound addition, displayed via a signal collection and analysis system (BL-420E + Systerm, Chengdu TME Technology Co., Ltd., Chengdu, China). When the current changed to the 1 mV pulse, it was used to estimate transepithelial resistance according to Ohm’s law [40]. Ten minutes after the Isc plateau, ATP was applied to the apical side, and then the ATP-induced Isc was recorded.

In the Cl^–^-free BRS, NaCl, KCl, and CaCl_2_ were replaced by the respective salts of gluconate. In the HCO_3_^–^-free solution, HCO_3_^−^ was substituted by 10 mM HEPES and bathed with 100% O_2_ to maintain pH at 7.4. In the Cl^–^- and HCO_3_^–^-free solution, the components were the same as in the Cl^−^-free solution, except that NaHCO_3_ was replaced by 20 mM Na-gluconate and 10 mM HEPES. Inhibitors or a vehicle (0.5% DMSO, D2650, Sigma, Burlington, MA, USA) were added to the buffer medium for 20 min before ATP was applied to the apical side. The dosages of inhibitors were as follows: 100 uM U73122 (U6756, Sigma, USA), a phospholipase C (PLC) inhibitor; 100 uM 2-APB (100065, Sigma, USA), an inositol (1,4,5) trisphosphate (IP3) receptor (IP3R) inhibitor; 100 uM BAPTA/AM (A1076, Sigma, USA), a Ca^2+^ chelator; 40 uM T16Ainh-A01 (SML0493, Sigma, USA), a specific CaCC inhibitor; and 40 uM CFTR (inh)-172 (C2992, Sigma, USA), a specific CFTR inhibitor; 10uM H_89_ (B1427, Sigma, USA), a specific cAMP-dependent protein kinase (PKA) inhibitor; and 30 uM MDL12330A (M182, Sigma, USA), an adenylate cyclase (AC) inhibitor. Different inhibitors were individually used in the conjunctival samples.

### 4.5. Measurement of Intracellular cAMP

The concentration of intracellular cAMP in the conjunctival epithelium was measured using a cAMP assays kit (E-EL-0056c, Elabscience Biotechnology Co., Ltd., Wuhan, China) according to the manufacturer’s protocol. Optical density readings were measured using a SynergyTM H1 multifunctional microplate reader (BioTek Instruments, Inc., Winooski, VT, USA). Concentrations were determined using a standard curve as a reference.

### 4.6. Measurement of Intracellular Ca^2+^

Intracellular Ca^2+^ was measured using the fluorescence dye Fluo-3/AM according to a modified procedure described in a previous report [41]. The conjunctival epithelium from the rabbits was digested and harvested with 1 mg/mL dispase-II solution for approximately 1 h, seeded in DMEM/F12 medium on glass-bottom cell culture dishes (20 mm, NEST, Wuxi, China), and cultured at 37 °C. After 10 min of cell adhesion, the cells were prepared and loaded with the fluorescent dye Fluo-3/AM (Molecular Probes, Eugene, OR, USA). The fluorescence signal was monitored and recorded using a laser-scanning confocal imaging system (TCS SP2; Leica Microsystems, Wetzlar, Germany). The change in fluorescence intensity after ATP treatment was normalized to the initial intensity.

### 4.7. Statistical Analysis

All statistical analyses were performed using SPSS version 22.0. All results were expressed as mean ± SEM. Statistical significances were determined using a Student’s t-test (two groups), and a repeated measures ANOVA, followed by a Least significant difference post hoc test (groups with repeated measurements). The differences between the groups were considered statistically significant if *p* was less than 0.05.

## 5. Conclusions

The results of this study indicate that the Ca^2+^-activated CaCC-CFTR pathway plays a key role in conjunctival secretion. The impairment of the P2Y_2_R-initiated CaCC-CFTR may be a key mechanism of the conjunctival secretion dysfunction in eDED.

## Figures and Tables

**Figure 1 ijms-23-14399-f001:**
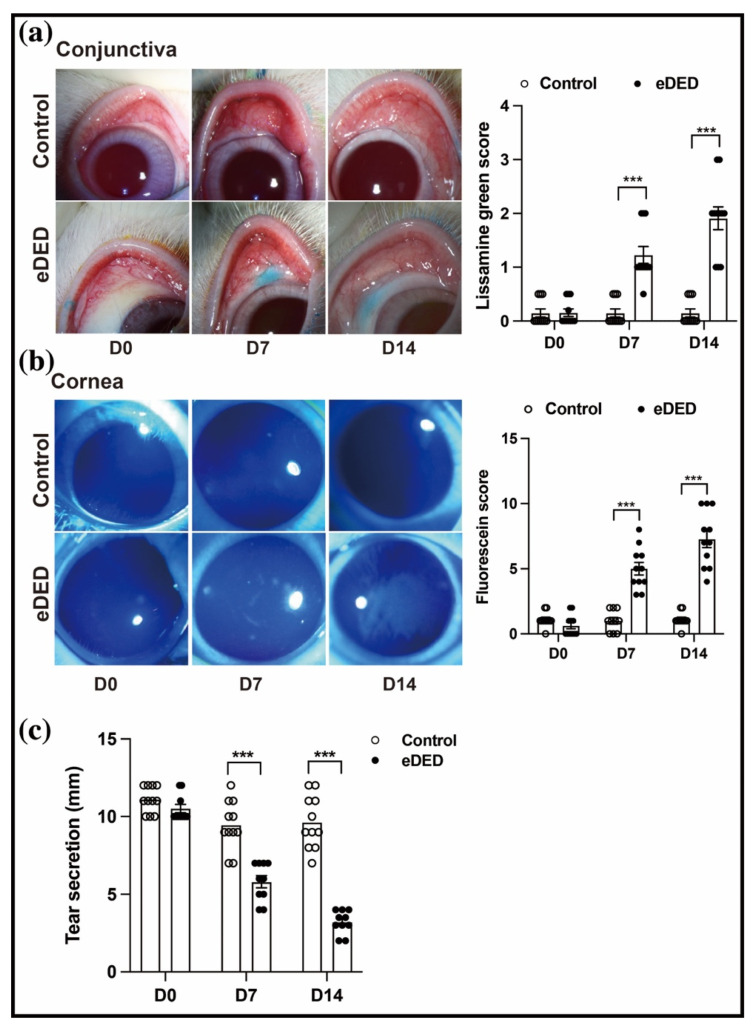
Ocular surface injury and tear secretion decreased in eDED. (**a**) Left, representative images of conjunctiva lissamine green staining in the control and eDED groups. Right, bar graph showing the score of conjunctiva lissamine green in the control (n = 10) and eDED groups (n = 11). (**b**) Left, representative images of corneal fluorescein staining in the control and eDED groups. Right, bar graph showing the score of cornea fluorescein in the control (n = 10) and eDED groups (n = 11). (**c**) Bar graph showing the tear volume in the control (n = 11) and eDED groups (n = 10). Data are represented as mean ± SEM. *** *p* < 0.001.

**Figure 2 ijms-23-14399-f002:**
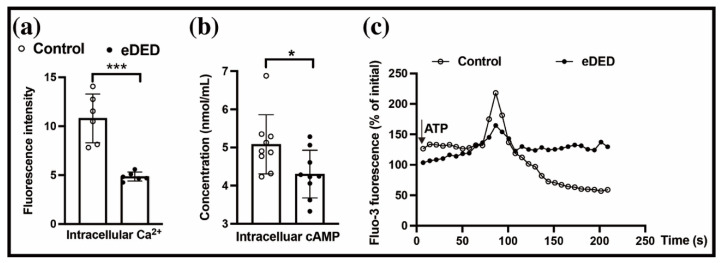
Ca^2+^ and cAMP deficiencies in the conjunctiva epithelium of eDED. (**a**) Bar graph showing intracellular Ca^2+^ of conjunctiva epithelium in the control and eDED groups (n = 6 in each group). (**b**) Bar graph showing the intracellular cAMP of the conjunctiva epithelium in the control and eDED groups (n = 9 in each group). (**c**) Line graph showing the intracellular Ca^2+^ of conjunctiva epithelium after ATP treatment in the control and eDED groups (n = 6 in each group). Data are represented as mean ± SEM. * *p* < 0.05, *** *p* < 0.001.

**Figure 3 ijms-23-14399-f003:**
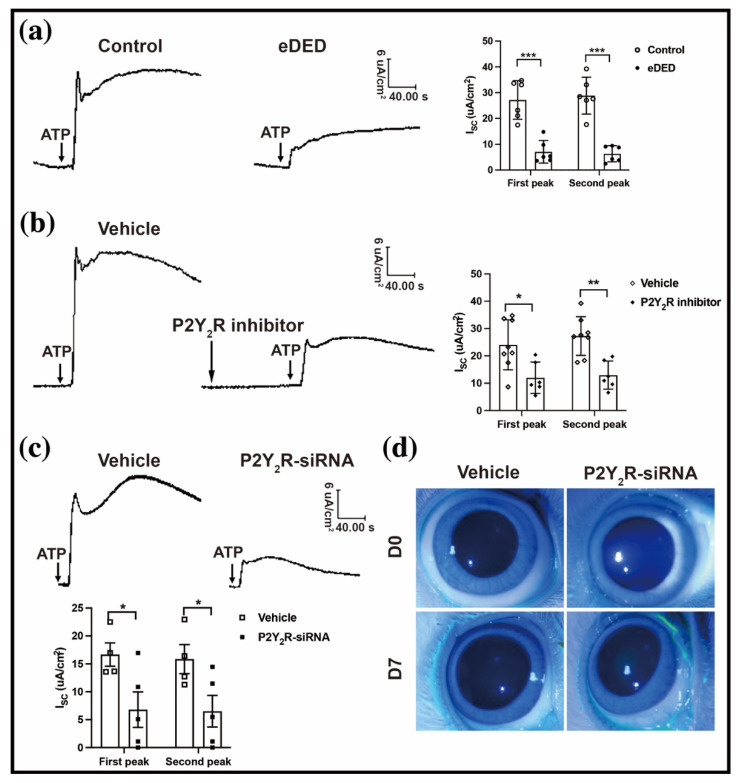
P2Y_2_R is indispensable for conjunctival fluid secretion. (**a**) Left, representative images of ATP-induced conjunctival Isc in the control and eDED groups; right, bar graph showing the first and second peaks of ATP-induced conjunctival Isc in the control and eDED groups, n = 6 in each group. (**b**) Left, representative images of ATP-induced conjunctival Isc of the vehicle and P2Y_2_R antagonist (suramin) groups; right, bar graph showing the first and second peaks of ATP-induced conjunctival Isc in the vehicle (n = 8) and P2Y_2_R antagonist groups (n = 6). (**c**) Top, representative images of ATP-induced conjunctival Isc in the vehicle and P2Y_2_R-siRNA groups. Bottom, bar graph showing the first and second peak of ATP-induced conjunctival Isc in the vehicle (n = 4) and P2Y_2_R-siRNA group (n = 5). (**d**) Representative images of cornea fluorescein in the vehicle and P2Y_2_R-siRNA group. Data are represented as mean ± SEM. * *p* < 0.05, ** *p* < 0.01, *** *p* < 0.001.

**Figure 4 ijms-23-14399-f004:**
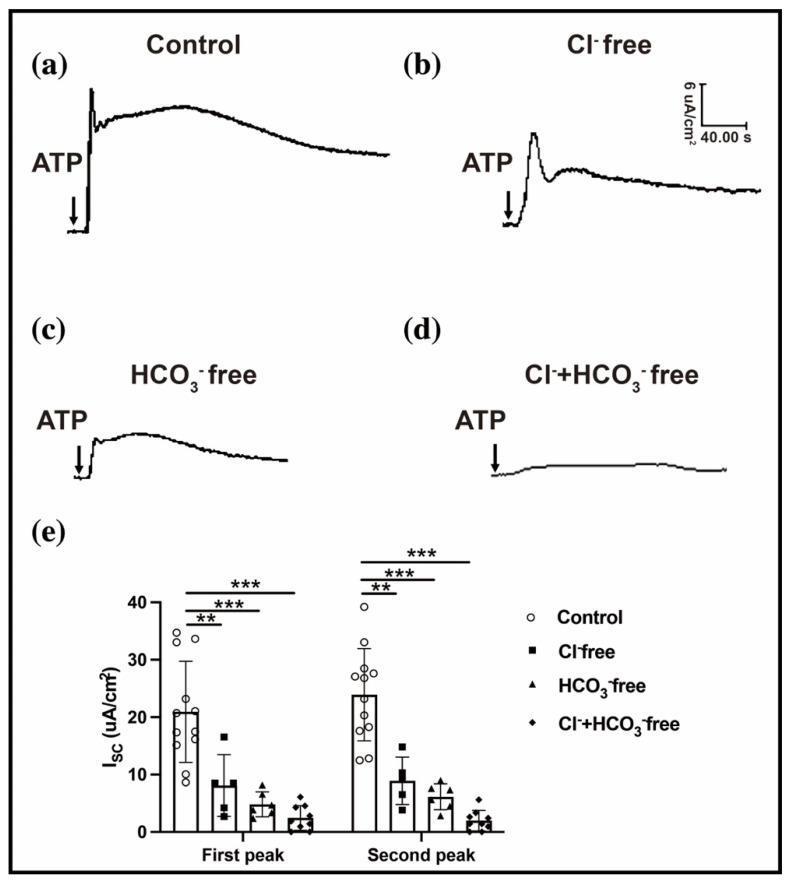
ATP-induced conjunctival Isc related to Cl^–^ channel. (**a**–**d**) Representative image of ATP-induced conjunctival Isc in the control group (**a**), Cl^–^-free group (**b**), HCO_3_^–^-free group (**c**), and both Cl^–^-free and HCO_3_^–^-free groups (**d**). (**e**) Bar graph showing the first peak and second peaks of ATP-induced conjunctival Isc in the control (n = 12), Cl^–^-free (n = 5), HCO_3_^–^-free (n = 6), and both Cl^–^-free and HCO_3_^–^-free groups (n = 9). Data are represented as mean ± SEM. All data were compared with the control group. ** *p* < 0.01, *** *p* < 0.001.

**Figure 5 ijms-23-14399-f005:**
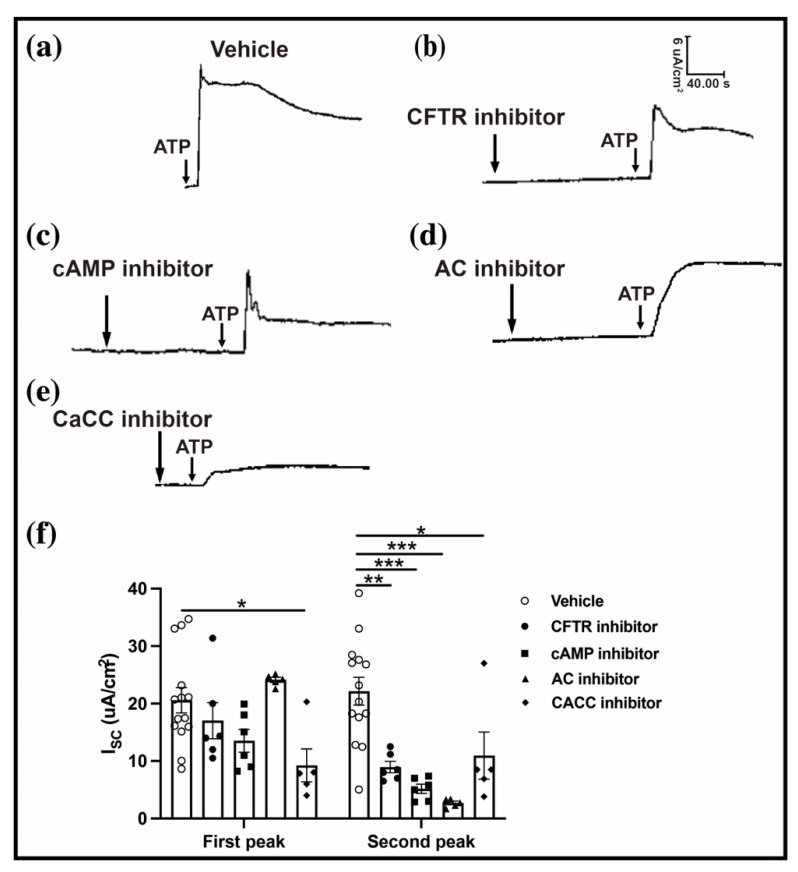
CaCC-CFTR pathway related to ATP-induced conjunctival Isc. (**a**) Representative image of ATP-induced conjunctival Isc in the vehicle group. (**b**–**e**) Representative image of ATP-induced conjunctival Isc using a special CFTR inhibitor (CFTR_inh−172_), a special cAMP blocker (H_89_), an AC inhibitor (MDL 12330A), and a special CaCC inhibitor (T16A_inh−A01_). (**f**) Bar graph showing the first and second peaks of ATP-induced conjunctival Isc in the vehicle group (n = 14), CFTR special inhibitor group (n = 6), AC inhibitor group (n = 5), and CaCC inhibitor group (n = 5). Data are represented as mean ± SEM. CFTR, cystic fibrosis transmembrane conductance regulator; AC, adenylyl cyclase; CaCC, Ca^2+^-activated Cl^−^ channel. All results were compared with those of the control group. * *p* < 0.05, ** *p* < 0.01, *** *p* < 0.001.

**Figure 6 ijms-23-14399-f006:**
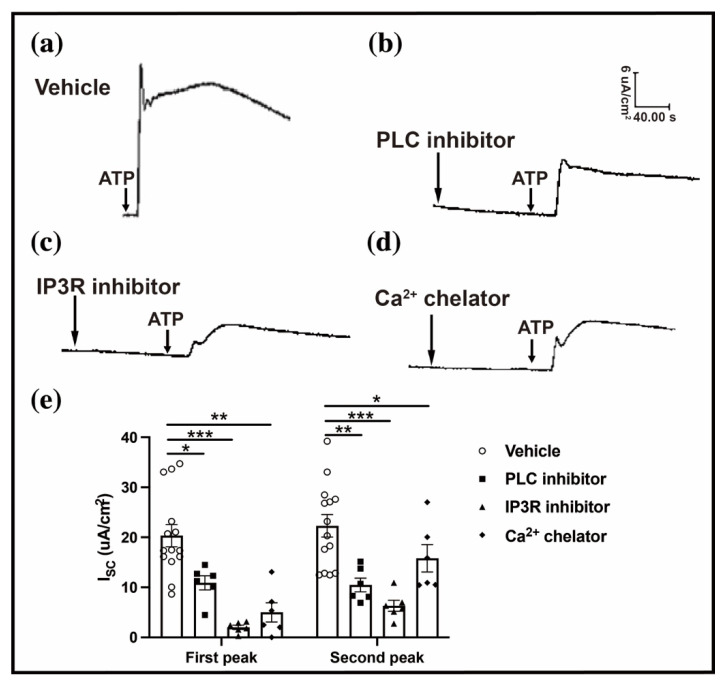
Ca^2+^ participates in ATP-induced conjunctival Isc. (**a**) Representative image of ATP-induced conjunctival Isc in the vehicle group. (**b**–**d**) Representative image of ATP-induced conjunctival Isc in the PLC inhibitor (U73122) group, IP3R inhibitor (2-APB) group, and Ca^2+^ chelator (BAPTA/AM) group. (**e**) Bar graph showing the first and second peaks of ATP-induced conjunctival Isc in the vehicle group (n = 14), PLC inhibitor group, IP3R inhibitor group, and Ca^2+^ chelator group (n = 6 in each group). Data are represented as mean ± SEM. PLC, phospholipase C; IP3R, inositol (1,4,5) trisphosphate receptor. All results were compared with those of the control group. * *p* < 0.05, ** *p* < 0.01, *** *p* < 0.001.

## Data Availability

Not applicable.

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
