# Peer review of "Conjunctival Fluid Secretion Impairment via CaCC-CFTR Dysfunction Is the Key Mechanism in Environmental Dry Eye"

_ijms, 2022, doi:10.3390/ijms232214399_

Round 1

Reviewer 1 Report

The manuscript by Zhang et al. investigates the role of P2Y2R dysfunction on eDED. Overall the study is difficult to understand due to many grammar errors and the manuscript would benefit from editing. There is also a lack of detail and rigor in the Results section. For instance, a description of the fluorescence staining method in the Results would make the data easier to understand. Also the efficiency of the siRNA knockdown is not reported, and the control that was used is a saline solution instead of a siRNA with a random sequence, making the experiment difficult to interpret. Further the authors report values with overlapping error bars for Figure 2b and I am unsure whether the reported difference is significant.

Reviewer 2 Report

First of all, I would like to congratulate the authors for the work. The point of view is original and its reading has been interesting.

Despite this, perhaps because of my speciality, I found the structure of the work curious. The fact that the "Materials and Methods" section comes after the "results" and "discussion" sections has been somewhat confusing to me.

However, the research sheds light on a not fully understood dysfunction that causes serious visual problems for many people. The findings found could represent a new way of addressing this dysfunction.

I would encourage them to continue advancing in their research in order to offer clinicians more tools to deal with this pathology.
